# Do Vision Encoders Truly Explain Object Hallucination?: Mitigating Object Hallucination via Simple Fine-Grained CLIPScore

**Hongseok Oh**                                                                          *cxv0519@uos.ac.kr*
*Department of Artificial intelligence*
*University of Seoul*

**Wonseok Hwang**                                                                *wonseok.hwang@uos.ac.kr*
*Department of Artificial intelligence*
*University of Seoul*

**Reviewed on OpenReview:** *https://openreview.net/forum?id=JTua6tDPgZ*

## Abstract

Recently, Large Vision-Language Models (LVLMs) show remarkable performance across various domains. However, these models suffer from object hallucination. In this work, we study object hallucination primarily in a discriminative, retrieval-style evaluation setting (OHD-Caps), rather than in free-form caption generation. This study revisits the previous claim that the cause of such hallucinations lies in the limited representational capacity of the vision encoder. Our analysis implies that the capacity of the vision encoder is not necessarily a major limiting factor in detecting object hallucination. Based on this insight, we propose Fine-grained CLIPScore (F-CLIPScore), a simple yet effective evaluation metric that enhances object-level granularity by incorporating text embeddings at the noun level. Evaluations on the OHD-Caps benchmark show that F-CLIPScore significantly outperforms conventional CLIPScore in accuracy by a large margin of **39.6%** without additional training. We further demonstrate that F-CLIPScore-based data filtering reduces object hallucination in LVLM (4.9% in POPE accuracy after alignment pretraining). Our code is publicly available at `https://github.com/abzb1/f-clip`

## 1 Introduction

Recent studies identify Large Vision-Language Models (LVLMs) as a leading approach for vision-language integration (Liu et al., 2023; Wang et al., 2024a; Zhu et al., 2023; Chen et al., 2024). However, like hallucinations in large language models (LLMs) (Ji et al., 2023; Zhang et al., 2023), LVLMs exhibit object hallucination, referring to nonexistent or misidentified objects that may undermine their reliability (Li et al., 2023a; Liu et al., 2024b). While object hallucination is often discussed in the context of caption generation, this paper focuses on discriminative vision–language models and retrieval-style evaluation protocols, where the goal is to select or score candidate captions given an image.

Liu et al. (2024c) built OHD-Caps, a dataset designed to measure object hallucination in a discriminative caption-selection setting. The dataset comprises 1.5k image-captions pairs where a model needs to select the best caption that does not show hallucinations. They found that CLIPScore (Hessel et al., 2021) achieved only 10–20% accuracy, and the further fine-tuning with the proposed objective function with the dataset improves the accuracy up to 80–90%. However, when they connected the OHD-Caps-trained CLIP to an LVLM and conducted full fine-tuning, the resulting accuracy sometimes drops, showing lower performance compared to original CLIP (for instance, 1st row of Table 4 in (Liu et al., 2024c) shows the accuracy in POPE benchmarks (Li et al., 2023b) drops from 85.4% to 81.2%).

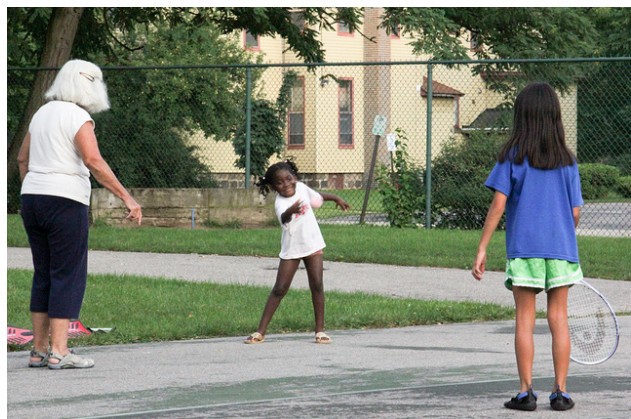

**CLIPScore** (**w/o training**): `A lady and two children in the street playing with a tennis racquet, a car nearby, and a chair.`

**CLIPScore** (**trained**): `A lady and two dogs in the park playing with a frisbee.`

**F-CLIPScore** (**w/o training**): `A lady and two children in the street playing with a tennis racquet.`

Figure 1: A representative example from the OHD-Caps test set is shown. The original CLIP selects a sentence mentioning "children" and "tennis" but adds hallucinated objects. The OHD-Caps-trained CLIP hallucinates "dog" and "frisbee" without introducing new content. In contrast, F-CLIPScore selects a sentence that preserves the original meaning without hallucinations.

Moreover, we found that OHD-Caps-trained CLIP sometimes tends to hallucinate by replacing existing objects. When both CLIPScore and OHD-Caps-trained CLIP fail, but our Fine-grained CLIPScore (introduced in a later section) succeeds, a clear pattern emerges: CLIPScore adds nonexistent objects, whereas OHD-Caps trained CLIP replaces existing ones. This trend holds across COCO, Flickr30k, and NoCaps with rates of 56%, 58%, and 59%, respectively. A representative example is shown in Figure 1. This suggests that object hallucination may stem from factors beyond the vision encoder's capacity.

To address this issue, we introduce Fine-grained CLIPScore (F-CLIPScore), a novel image-text correlation metric. F-CLIPScore leverages a sentence parser like spaCy Honnibal et al. (2020) and the forward pass of a Vision-Language Model (VLM) like CLIP Radford et al. (2021), offering an efficient way to evaluate the vision encoder's representational capacity. Our experimental results show that applying F-CLIPScore to the OHD-Caps test set improves accuracy by **+39.6%** without additional training. This indicates that the limited capacity of the vision encoder may not be the primary cause of object hallucination. Additionally, we verify that using F-CLIPScore for pretraining data curation in LVLMs enables the training of models with reduced hallucination, even with significantly fewer data samples. Notably, in LVLM pretraining, data filtering alone improved POPE accuracy by 4.9% compared to the baseline.

This study offers the following key contributions:

- We introduce Fine-grained CLIPScore, a novel evaluation metric that relies solely on forward propagation.

- We provide evidence suggesting that, in discriminative evaluation, object hallucination is not primarily explained by the vision encoder's representational capacity alone.

- We demonstrate that F-CLIPScore enables more efficient LVLM training with reduced object hallucination through pretraining data curation.

Our code is available at `https://github.com/abzb1/f-clip`.

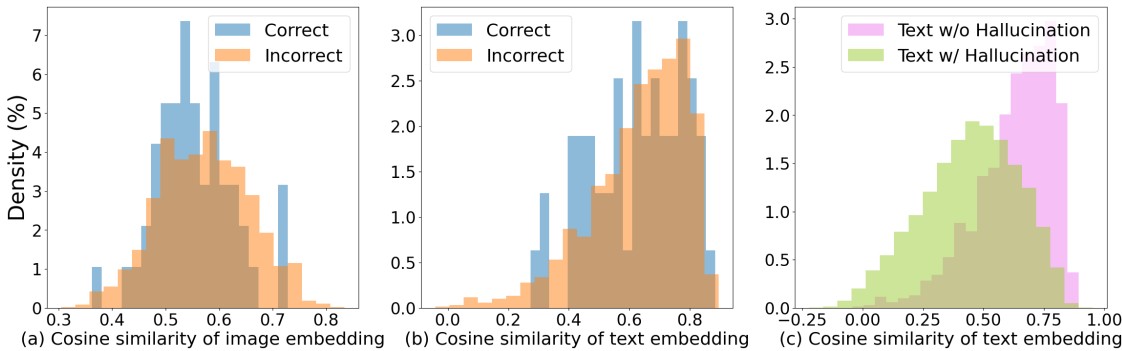

Figure 2: Histograms of cosine similarity between two embedding vectors: one from the original CLIP-L and the other from the OHD-Caps-trained CLIP-L. (a) The histogram from the vision encoders. `Correct` (blue) indicates the scores are from the examples where original CLIP-L predict the ground truth. The other examples are colored in orange. (b) The histogram from the text encoders. Same color scheme is employed. Measured only on ground truth text. (c) The cosine similarity distribution between text embeddings of text without object hallucination (purple) and with object hallucination text (green) for all samples.

## 2 Related Work

Object hallucination refers to cases in which the generated textual descriptions include objects that do not correspond to the given image (Liu et al., 2024b). LVLMs generally consist of three components: a vision encoder, an LLM, and an adapter (Liu et al., 2023). The structural characteristics of LVLMs contribute to object hallucination, which arises from multiple intertwined factors (Liu et al., 2024b). While some studies argue that hallucinations can be mitigated by enhancing the decoding process of the LLM (Manevich & Tsarfaty, 2024; Wang et al., 2024b), others suggest that one of the causes lies in the limited representational capacity of the vision encoder (Liu et al., 2024c). Additionally, some research indicates that training the adapter with contrastive data is essential to reduce object hallucination (Jiang et al., 2024). Moreover, the trained bias of the model has also been identified as a cause of hallucination (Hu et al., 2023; Liu et al., 2024a).

CLIPScore (Hessel et al., 2021) is a reference-free evaluation metric that assesses the consistency between an image and text caption by computing the cosine similarity between the embeddings generated by the vision encoder and text encoder of the CLIP model. Beyond its application in measuring caption quality, several studies have also leveraged CLIPScore for data curation in the training of Vision-Language Models (VLMs) (Schuhmann et al., 2021; Gadre et al., 2023).

A recent study utilized CLIPScore to evaluate object hallucination in Vision-Language Models (VLMs) (Liu et al., 2024c). Their findings suggest that this phenomenon stems from the limited capacity of the vision encoder. In this study, we carefully reassess this claim and demonstrate that object hallucination is not necessarily caused by the limitations of the vision encoder alone.

## 3 Methods

### 3.1 Motivation

Based on our initial observation that OHD-Caps-trained CLIP occasionally does not yield better results (Section 1), we further investigate how fine-tuning affects the embedding vectors produced by vision and text encoders of CLIP. We compute the cosine similarity between two embedding vectors: one from the original CLIP-L and the other from the OHD-Caps-trained CLIP-L, using image-text pairs in the OHD-Caps test set.

We first computed the distribution of cosine similarity between the image embeddings from CLIP-L and OHD-Caps-trained CLIP-L, focusing on the samples that were correctly predicted by CLIP-L. We then performed the same analysis on the samples that were incorrectly predicted by CLIP-L and compared the

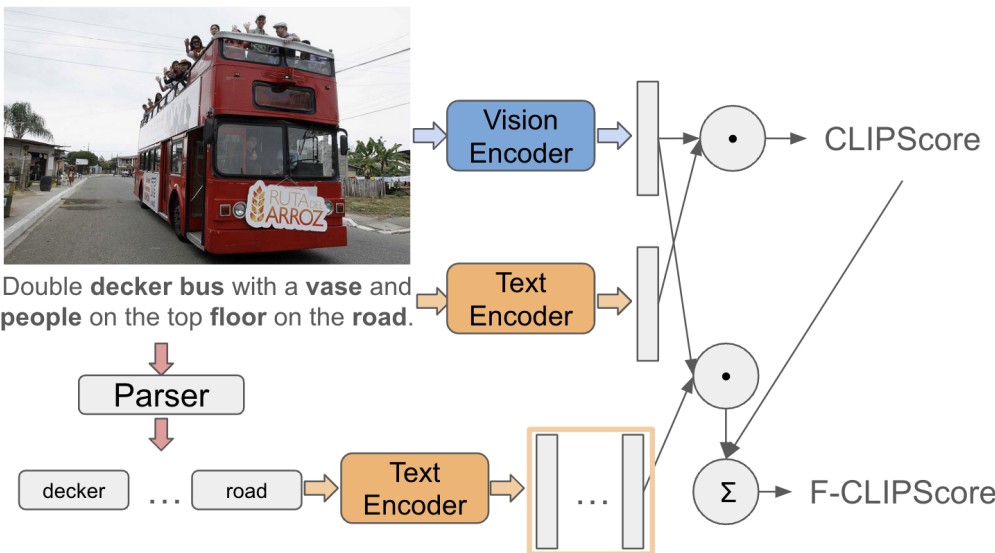

Figure 3: The graphical representation of F-CLIPScore.

two distributions. As shown in Figure 2a, we observe that there is no substantial difference between the two distributions. We conducted the same analysis using ground-truth text embeddings of the pre-trained and fine-tuned models, and similarly found no significant difference in the cosine similarity distributions between the samples that were correctly and incorrectly predicted by CLIP-L (Figure 2b). Two-sample t-tests yield p-values of 0.11 for a and 0.67 for b. In contrast, Figure 2c reveals significant changes in text embeddings for captions with hallucination (green), and without hallucination (purple) highlighting the distinct adaptation of text representation. Two-sample t-tests yield near zero p-value for c. These observations suggest that OHD-Caps training induces more noticeable changes in the text representation space, particularly in discriminating hallucinated from non-hallucinated captions, whereas the vision representation space exhibits relatively minor changes.

### 3.2 Fine-grained CLIPScore

Motivated by these observations, we propose a simple metric called Fine-grained CLIPScore (F-CLIPScore), which enhances the discriminative power of the VLM in order to utilize textual information with more granularity without additional training. F-CLIPScore first utilizes the spaCy parser Honnibal et al. (2020) to extract nouns from a given sentence. Then, it evaluates the quality of an image caption by averaging the CLIPScore of the entire sentence and each individual noun (Figure 3).

Mathematically, given an image $I$ and a caption $C$ containing a total of $N$ nouns, each denoted as $n_i$, F-CLIPScore is defined as Eq. 1.

$$\text{F-CLIPScore}(I, C) = \frac{\text{CLIPScore}(I, C) + \sum_{i=1}^{N} \text{CLIPScore}(I, n_i)}{N + 1} \tag{1}$$

## 4 Experiments

For evaluating OHD-Caps test set, F-CLIPScore only requires an additional parsing step (we used en_core_web_sm of spaCy (Honnibal et al., 2020)), which takes on average 170 ms per sentence and can be parallelized across samples. We note that this parser runs in linear time with respect to the length of the sentence (Honnibal & Johnson, 2015). For training CLIP with Eq. 2 on the OHD-Caps train set, we used a batch size of 64 and a learning rate of 1e-5, which required 3 hours on an H100 GPU. For LLaVA

| OHD-Caps ACC (%, ↑) | | | |
|---|---|---|---|
| **Metric** | **COCO** | **Flickr30k** | **NoCaps** |
| w/o training | | | |
| CLIPScore | 22.6 | 22.6 | 12.4 |
| F-CLIPScore | **62.2** | **62.2** | **46.6** |
| trained w/ OHD-loss[†] | | | |
| CLIPScore | 79.8 $_{\pm 1.7}$ | **84.8** $_{\pm 1.6}$ | 84.0 $_{\pm 0.7}$ |
| trained w/ F-CLIPScore loss | | | |
| CLIPScore | **80.5** $_{\pm 1.8}$ | **84.8** $_{\pm 1.3}$ | **84.1** $_{\pm 1.3}$ |

†: We trained openai/clip-vit-large-patch14 with the train code from Liu et al. (2024c)

Table 1: Accuracy on the OHD-Caps test set evaluated with OpenAI CLIP-L. For trained models, we report the average evaluation results over 10 runs with different random seeds.

pretraining, we employed an effective batch size of 256 and a learning rate of 1e-3, which took 7 hours on two H100 GPUs.

## 5 Results

By utilizing the F-CLIPScore, which directly leverages the image embeddings from the CLIP vision encoder without any training or gradients, while only adding a parsing step during forward propagation, we can efficiently gain insights into whether the issue of object hallucination arises from the limited capability of the vision encoder.

### 5.1 F-CLIPScore on OHD-Caps

We evaluated the OHD-Caps test set, an object hallucination assessment dataset, using the proposed F-CLIPScore. As shown in Table 1, evaluation results with OpenAI CLIP ViT-L Radford et al. (2021) indicate that F-CLIPScore outperformed the baseline model by up to 39.6% without additional training (row 4 vs. 5). However, it still performed 17.6% to 37.4% worse than trained models (row 5 vs. 7). This may indicate that although CLIP vision encoders may not be the main cause of object hallucination, further training may enhance their capability.

To test whether F-CLIPScore is orthogonal to the previously proposed method Liu et al. (2024c), we modify the loss as

$$L_{CLIP} + L_{OHD} + \frac{\alpha}{B} \sum_{i=1}^{B} (1 - \text{F-CLIPScore}(I_i, C_i)) \tag{2}$$

where $L_{CLIP}$ is the contrastive loss proposed in Radford et al. (2021), and $L_{OHD}$ is the marginal loss proposed in Liu et al. (2024c). $B$ denotes the batch size, while $I_i$ and $C_i$ are image and caption of the $i$-th positive pair. $\alpha$ is a hyperparameter that we set to 0.3. Applying F-CLIPScore as a loss improved performance by 0.7 percentage points on COCO and 0.1 percentage points on NoCaps (row 7 vs. 9). These results suggest that F-CLIPScore could be a complementary component in VLM training.

We randomly replaced the caption embeddings into noun embeddings extracted from a Wikipedia corpus (Davies, 2015) to examine how F-CLIPScore responds to such perturbations. As shown in Table 2, we observed the expected decline in accuracy as nouns were replaced. This indicates that the strong performance of F-CLIPScore, even without additional training, is not due to random noise, but rather stems from its ability to more effectively leverage fine-grained textual embeddings.

We experimented with alternative configurations of F-CLIPScore using verbs and noun phrases instead of nouns. As shown in Table 3, using verbs yielded little to no improvement, while noun phrases performed better than verbs but still fell short of the performance achieved with nouns.

| Replacement Rate | | | | |
|---|---|---|---|---|
| 0.2 | 0.4 | 0.6 | 0.8 | 1.0 |
| **COCO Acc mean $\pm$ std (%, $\uparrow$)** | | | | |
| 38.9±2.5 | 35.9±1.4 | 31.0±1.7 | 23.5±2.0 | 19.9±1.3 |
| **Flickr30k Acc mean $\pm$ std (%, $\uparrow$)** | | | | |
| 37.5±1.5 | 33.0±1.2 | 26.2±2.1 | 22.0±1.6 | 18.9±3.0 |
| **NoCaps Acc mean $\pm$ std (%, $\uparrow$)** | | | | |
| 29.4±1.9 | 24.2±2.3 | 21.2±2.0 | 16.2±1.7 | 16.5±0.4 |

Table 2: F-CLIPScore results on the OHD-Caps test set with different random noun replacement rates (0.2 to 1.0). Each value shows the mean accuracy $\pm$ standard deviation over five random seeds.

| OHD-Caps ACC (%, $\uparrow$) | | | |
|---|---|---|---|
| **Metric** | **coco** | **flickr30k** | **nocaps** |
| CLIPScore | 22.6 | 22.6 | 12.4 |
| F-CLIPScore$_V$ | 23.6 | 24.8 | 15.8 |
| F-CLIPScore$_{NP}$ | 54.2 | 57.8 | 39.6 |
| F-CLIPScore$_N$ | **62.2** | **62.2** | **46.6** |

Table 3: Accuracy on the OHD-Caps test set evaluated with OpenAI CLIP-L. F-CLIPScore$_N$ refers to the noun-based F-CLIPScore defined in Eq. 1. F-CLIPScore$_{NP}$ uses noun phrases instead of individual nouns, and F-CLIPScore$_V$ replaces the nouns with verbs.

Our F-CLIPScore even outperforms the training-based methods (Zhang et al., 2024; Yuksekgonul et al., 2023), as shown in Table 4.

Furthermore, Table 5 demonstrates that F-CLIPScore consistently achieves gains across different scales and architectures of vision-language models.

## 5.2 LVLM Pretrain Data Curation with F-CLIPScore

As shown in Section 5.1, F-CLIPScore was able to exhibit competent performance in detecting object hallucination without any training, and could further serve as an complementary method for training VLMs. We aimed to investigate whether F-CLIPScore can influence the pretraining process of LVLM by acting as a data filter between the vision encoder and the LLM. To this end, we applied F-CLIPScore to filter the pretraining data for LLaVA (Liu et al., 2023), which consists of 558k samples, by removing the bottom $x\%$ of the alignment training set based on F-CLIPScore.

As shown in Table 6, training the alignment model on the top 70% of data curated by F-CLIPScore resulted in a +4.9% improvement in POPE accuracy compared to training on the entire dataset (row 5 vs. 7). In contrast, using the OHD-Caps-trained CLIP for filtering yielded only marginal gains, and random filtering showed no improvement. These results may suggest that F-CLIPScore effectively captures object hallucination-related

| **OHD-Caps Accuracy** | **COCO** | **Flickr30k** | **NoCaps** |
|---|---|---|---|
| CLIP-B/32 (CLIPScore) | 15.2 | 17.6 | 10.2 |
| CECLIP (Zhang et al., 2024) | 32.8 | 28.0 | 25.0 |
| NegCLIP (Yuksekgonul et al., 2023) | 52.8 | 40.8 | 23.4 |
| **CLIP-B/32 (F-CLIPScore)** | **56.0** | **53.2** | **43.0** |

Table 4: OHD-Caps accuracy across different methods.

| OHD-Caps Accuracy | COCO | Flickr30k | NoCaps |
|---|---|---|---|
| SigLIP ViT-L (CLIPScore) (Zhai et al., 2023) | 48.6 | 38.4 | 30.2 |
| **SigLIP ViT-L (F-CLIPScore)** | **69.0** | **64.2** | **54.2** |
| EVA-CLIP ViT-L (CLIPScore) (Sun et al., 2023) | 38.6 | 31.8 | 22.6 |
| **EVA-CLIP ViT-L (F-CLIPScore)** | **69.6** | **64.8** | **55.4** |
| CLIP ViT-B (CLIPScore) | 15.2 | 17.6 | 10.2 |
| **CLIP ViT-B (F-CLIPScore)** | **56.0** | **53.2** | **43.0** |
| CLIP ViT-L (CLIPScore) | 22.6 | 22.6 | 12.4 |
| **CLIP ViT-L (F-CLIPScore)** | **62.2** | **62.2** | **46.6** |
| CLIP ViT-H (CLIPScore) | 36.8 | 31.4 | 20.6 |
| **CLIP ViT-H (F-CLIPScore)** | **57.6** | **59.4** | **46.6** |

Table 5: OHD-Caps accuracy across different vision-encoder architectures and sizes.

| POPE Acc ($\uparrow$, %) | Filtering rate (%) | | | | |
|---|---|---|---|---|---|
| **Filtering Method** | **20** | **30** | **40** | **50** | **60** |
| Random | 50.7 | 49.9 | 49.9 | 50.5 | 49.8 |
| CLIPScore (w/o train) | 52.2 | 47.3 | 50.0 | 53.1 | 50.0 |
| F-CLIPScore (w/o train) | 51.6 | **55.5** | 50.6 | 50.1 | 51.8 |
| CLIPScore (trained) | 49.8 | 49.8 | 50.8 | 49.8 | 52.6 |
| w/o filtering | 50.6 | | | | |

Table 6: Accuracy on the POPE benchmark after LLaVA pretraining (vision-text alignment training before SFT). We use CLIP-L Radford et al. (2021) as a vision encoder, and Llama 2 7B Touvron et al. (2023) as an LLM backbone. "trained" indicates the OHD-Caps-trained CLIP-L. "Random" refers to the removal of $x$% of samples chosen at random.

quality, even in general-purpose datasets. This finding underscores the need to explore alternative causes of object hallucination beyond the capacity of the vision encoder. Although model performance shows a non-linear trend with respect to the filtering ratio, our experiments used a fixed number of training epochs, which may account for this behavior. As Goyal et al. (2024) highlights the need to balance data quality with computational cost, identifying the optimal filtering ratio under such trade-offs remains an important direction for future research.

## 6 Conclusion

We introduce F-CLIPScore, a simple yet effective metric for evaluating fine-grained image-caption alignment and addressing object hallucination in Vision-Language Models. Unlike conventional CLIPScore, which relies solely on sentence-level embeddings, F-CLIPScore also incorporates noun-level embeddings. This refinement allows the model to better mitigate object hallucination without requiring additional training for the vision encoder. We validate F-CLIPScore by showing a +39.6% accuracy in OHD-Caps benchmark. We also show that data filtering based on F-CLIPScore can enhance LVLM performance in hallucination mitigation, even with a reduced dataset. Our results suggest that the limitations of existing evaluation metrics, rather than the vision encoder itself, may contribute to object hallucination because they fail to accurately reflect the vision encoder's true capacity.

## Limitations

First, our study primarily targets discriminative evaluation and retrieval-style scoring (e.g., OHD-Caps, POPE) and does not directly evaluate hallucination in free-form caption generation. Investigating how noun-level scoring can be integrated into generative decoding or training objectives remains future work.

While this study proposes a method for analyzing and mitigating object hallucination using F-CLIPScore, it is subject to the following limitations. First, we were unable to conduct experiments on the Supervised Fine-Tuning (SFT) for Large Vision-Language Models (LVLMs). In the LVLM training pipeline, after the alignment pretraining phase—where the vision encoder and LLM remain frozen—the SFT stage follows, in which these components are unfrozen and further trained. However, due to computational resource limitations, we did not fully explore the potential impact of F-CLIPScore during SFT. Future work could investigate ways to incorporate F-CLIPScore into the SFT process to enhance training effectiveness.

Second, our method faces linguistic constraints and challenges in multilingual generalization. This study employs the spaCy parser (Honnibal et al., 2020) to extract nouns from text, a technique that performs relatively reliably in well-structured languages such as English. However, parsing accuracy may vary across different languages, potentially leading to inconsistencies in F-CLIPScore computation. To address this, future research should explore the scalability of F-CLIPScore by evaluating its effectiveness on multilingual datasets and refining the parsing methodology for broader linguistic applicability.

**Acknowledgments**

This work was supported by the National Research Foundation of Korea (NRF) grant funded by the Korea government(MSIT) (RS-2025-23524855).

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

# A    Qualitative Analysis of F-CLIPScore Filtering

We first observed the degree of overlap between the F-CLIPScore and CLIPScore metrics when filtering the data. At filtering rates of 20, 30, 40, 50, and 60, the proportions of overlapping image–caption pairs filtered by both metrics were 57%, 62%, 67%, 71%, and 76%, respectively. We then randomly sampled 10 image–caption pairs that did not overlap between the two metrics' filtered sets at the 30% filtering rate, which was the setting that yielded the best performance when using the F-CLIPScore metric. The results are presented in Figure 4. As shown in Figure 4, the images filtered by CLIPScore (upper row) appear to include some that should not have been filtered out, despite having normal and semantically correct captions. Although cases such as (g) and (h) involve repetitive lexical usage—where a word is repeated even if it correctly refers to an object present in the image—it is still reasonable for such captions to receive lower scores. However, it is unfortunate that these samples were filtered only by CLIPScore and not by F-CLIPScore.

On the other hand, the images filtered by F-CLIPScore (lower row in Figure 4) include cases such as (l), where even though there is repetitive lexical usage, the corresponding object in the image is actually a model rather than a designer, making the filtering reasonable. There were also examples like (n), where the captioned object is missing from the image, and (r), where the image shows a personal debt chart for "Brainy" rather than a "brain". These qualitative samples provide insights into how filtering with F-CLIPScore influenced the performance of the LVLMs.

Furthermore, we measured the CLIPScore and F-CLIPScore for each image–caption pair across the entire LLaVA-Pretrain dataset and sorted them in ascending order, such that a higher score corresponds to a higher rank. We then sorted the samples by the difference between the two ranks (F-CLIPScore rank − CLIPScore rank) and selected the top 10 samples with the largest positive values (i.e., those that CLIPScore rated higher than F-CLIPScore, shown in the upper row of Figure 5) and the bottom 10 samples with the largest negative values (i.e., those that F-CLIPScore rated higher than CLIPScore, shown in the lower row of Figure 5).

As shown in Figure 5, the samples on side (I), which CLIPScore rated higher than F-CLIPScore, contain many text-rendered images. Conversely, the samples on side (II), which F-CLIPScore rated higher than CLIPScore, tend to exhibit duplicated lexical usage. These findings are consistent with the previous observations, and such a trade-off is expected to make determining an appropriate filtering rate more challenging.

# B    Licenses for the datasets and models used

The LLaVA pre-training code is licensed under the Apache License, Version 2.0. LLaMA 2 7B is released under the LLaMA Community License. The LLaVA pretraining dataset is licensed under a combination of LAION, CC, and SBU licenses. The OpenAI CLIP model is distributed under the MIT License. COCO is available under the Creative Commons Attribution 4.0 License (CC BY 4.0). Flickr30k is provided under a custom license that permits use for non-commercial research and/or educational purposes. NoCaps is distributed under the Creative Commons Attribution 2.0 License (CC BY 2.0). We used the datasets and models in accordance with their respective licenses.

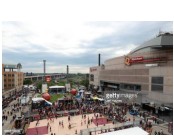

(a) a view of the crowd during the 2016 nba allstar game at madison arena on august 30 2018 in new

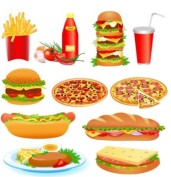

(b) a group of food vectors on a white background

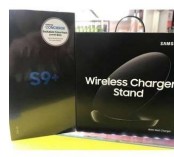

(c) a pair of samsung charger stand with charging and charger in the background

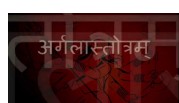

(d) an illustration of lord krishna in marathi language

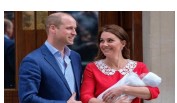

(e) the duke and duchess of cambridge pose for a family portrait

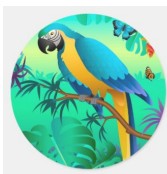

(f) blue and yellow macaw in the jungle sticker

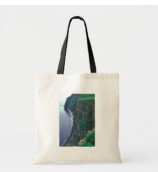

(g) a cliff cliff is on the island of maui canvas bag

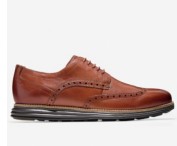

(h) the brown shoe is shown with two black soles and a black rubber outsole

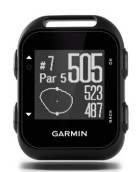

(i) garmin edge 20 gps watch with heart rate and gps coordinates

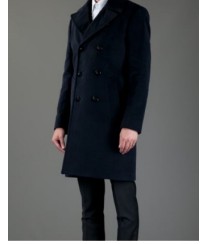

(j) ayrmoors - long length coat 8

(I) Image samples filtered by CLIPScore at the 30% filtering rate

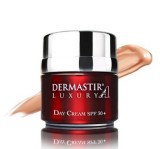

(k) dermastir'd® lux ultra k wax cream spf

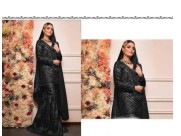

(l) an elegant black designer designer gown with beautiful lace work

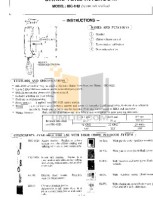

(m) pdf manual for microphone microphones for alto telephone

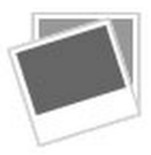

(n) a stainless steel sink with three drainers and two spout faucets

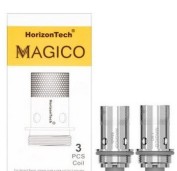

(o) horizon™ magico mesh coil

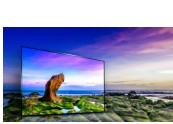

(p) tv on the stone beach on sunset stock photo

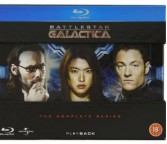

(q) battlestard galactica the complete series blu - ray

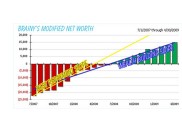

(r) an upward chart shows the number of complaints about brain disease

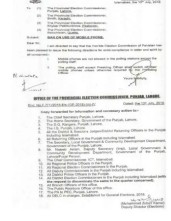

(s) the official documents of the commission for commissioning and remediarnation of power transmission units.png

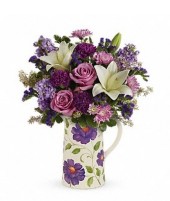

(t) telefish garden pitcher in new castle pa, the bouquet

(II) Image samples filtered by F-CLIPScore at the 30% filtering rate

Figure 4: Randomly sampled 10 image-caption pairs each from (I) samples filtered by CLIPScore and (II) samples filtered by F-CLIPScore at the 30% filtering rate. Images overlapping between the two metrics were excluded.

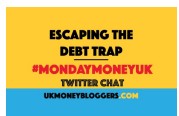

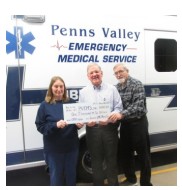

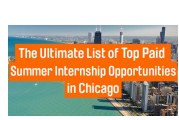

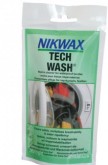

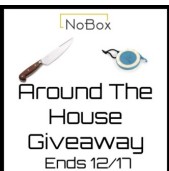

(a) the image shows a photo of a money trap with text escaping the debt trap mondaysmoneyuk twitter chat

(b) an elderly couple with a check presentation from the penns valley ambulance in michigan

(c) a lake michigan skyline and a chicago skyline with the text the ultimate list of top paid summer internship opportunities in

(d) nikwax tech wash for bike helmets, helmets and helmets

(e) a knife, a clochet, and a round-up of the nobox household appliances gift give

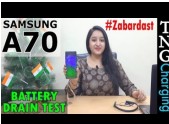

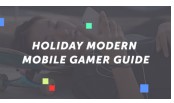

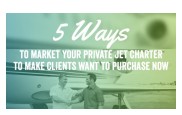

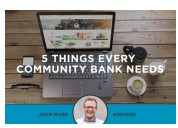

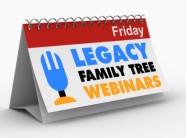

(f) samsung a70 battery drain test, testing and test battery at home in 4 days | best battery indicator

(g) a girl lying on a skateboard with the words holiday modern mobile game guide in front

(h) two men shaking hands outside of a plane with the title, 5 ways to market your private jet charter

(i) a laptop, phone, coffee cup, and keyboard with the words 5 things every community bank needs

(j) a calendar with the words legacy family tree webinars for a holiday meal

(I) Top 10 samples with the largest positive ranking differences

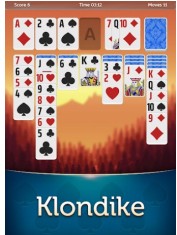

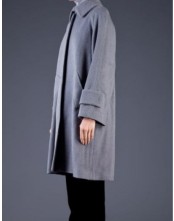

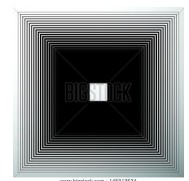

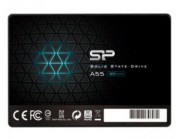

(k) solitaire solita solitaire solita solitaire solitaire solitaire solitaire solitaire solitaire sol

(l) alexander alexander - double breasted coat 9

(m) a screenplay screenplay with scriptwriting, handwritten

(n) an illusion illusion illusion illusion in black and white

(o) a52 250gb sat - 540 internal ssd

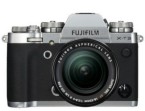

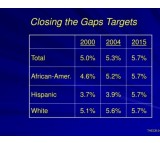

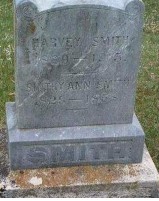

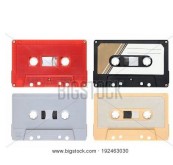

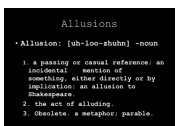

(p) a fujifilm m7 with the fuji 105mm f2

(q) the gap and the gap of the gaps gap of the gaps gap on the gaps gap of the gaps is

(r) the headstone of robert s dutton

(s) old cassette tape recorder

(t) the algorithm, with an allusion

(II) Bottom 10 samples with the largest negative ranking differences

Figure 5: The figure shows the top and bottom 10 samples from the entire LLaVA-Pretrain dataset, sorted by the difference between the F-CLIPScore rank and the CLIPScore rank. (I) represents samples that CLIPScore rated higher than F-CLIPScore, while (II) represents the opposite cases. Each caption is written below its corresponding image.

