# OpenReview forum: "Do Vision Encoders Truly Explain Object Hallucination?: Mitigating Object Hallucination via Simple Fine-Grained CLIPScore"
_TMLR — Accepted by TMLR_

### Review · Reviewer_44XK · 2025-09-30

**Summary Of Contributions:**

This paper challenges the prior claim that object hallucination in Large Vision-Language Models (LVLMs) primarily stems from the limited representational capacity of vision encoders. The authors introduce Fine-grained CLIPScore (F-CLIPScore), a simple, training-free evaluation metric that refines the standard CLIPScore by incorporating noun-level text embeddings (extracted via a parser like spaCy) alongside sentence-level embeddings. This enhances object-level granularity in assessing image-text alignment. Key contributions include:
(1) Empirical evidence from cosine similarity analyses and failure mode patterns showing that vision encoder limitations are not the main cause of hallucinations;
(2) Demonstrating that F-CLIPScore achieves a 39.6% accuracy improvement on the OHD-Caps benchmark without training, outperforming baselines and even some trained methods;
(3) Showing that using F-CLIPScore for pretraining data curation in LVLMs (e.g., LLaVA) reduces hallucinations by 4.9% on POPE, with fewer data samples.

**Audience:**

Yes

**Audience Explanation:**

TMLR's audience, which includes machine learning researchers focused on vision-language models, multimodal AI, and hallucination mitigation, would find this relevant. The paper addresses timely issues in LVLM reliability, such as object hallucinations, and offers practical tools for efficient training and evaluation. Findings on reevaluating vision encoder roles and data curation could influence work in scalable ML systems, aligning with TMLR's emphasis on foundational and applied research in areas like representation learning and model evaluation.

**Claims And Evidence:**

Yes

**Claims Explanation:**

The claims are well-supported by a combination of motivational analyses (e.g., cosine similarity distributions in Figure 2 showing minimal vision embedding changes post-fine-tuning) and rigorous experiments. Evaluations on OHD-Caps demonstrate clear accuracy gains (e.g., Table 1: 39.6% improvement over baseline CLIPScore), with multiple runs for variance and comparisons to trained models. Data filtering results on POPE (Table 2) convincingly show benefits like 4.9% accuracy uplift, isolated from confounders via random and alternative filtering baselines. Robustness checks in the appendix (e.g., Tables 3-6) further validate the metric across architectures (e.g., CLIP variants, SigLip, EVA-CLIP), though some experiments still need to be supplemented.

**Requested Changes:**

- Add ablation or comparative experiments to test scenarios involving the extraction of words with other properties (e.g., verbs or adjectives), random word extraction, or individual word extraction separately.
- Recommended: Add experiments or evidence to demonstrate that this noun-focused method does not affect the visual encoder's representation of other words and the overall sentence.
- Minor: Fix minor typos (e.g., the bracket error in the citation of the 4th paragraph in the Introduction section).

---

### Review · Reviewer_NwnV · 2025-10-09

**Summary Of Contributions:**

Strengths:
This paper proposes a novel approach to enhance the anti-hallucination capability of visual encoders without additional training, by observing and analyzing the cosine similarity between the trained CLIP in OHD-Caps and the original CLIP across different modalities. The method demonstrates effectiveness and provides insights into modality interactions.

Weaknesses:
However, the origin and derivation of the F-CLIPScore calculation formula are not thoroughly explained, which may hinder understanding and reproducibility. In addition, the experimental validation is limited, as comparisons are only performed on the OHD-Caps benchmark. This raises concerns regarding the generalizability of the proposed method. It would be beneficial to include more comprehensive experiments on additional benchmarks, such as LLaVA or other relevant datasets, to further substantiate the effectiveness and robustness of the approach.

**Additional Comments:**

None

**Audience:**

Yes

**Audience Explanation:**

The method proposed in this paper enables rapid assessment of image-text matching without the need for additional training, which indeed has certain practical applications.

**Broader Impact Concerns:**

This paper does not discuss the broader impact issues. It is recommended that the authors supplement the relevant content, including but not limited to: potential ethical risks arising from the application of the technology (such as privacy protection and algorithmic bias), societal impact (such as effects on vulnerable groups), legal compliance (such as whether data usage complies with laws and regulations), and environmental impact (such as energy consumption). Addressing these aspects will help improve the completeness and social responsibility of the paper.

**Claims And Evidence:**

No

**Claims Explanation:**

The motivation for proposing the F-CLIPScore calculation formula in this paper is not sufficiently clear, and the derivation process is not well articulated, making it difficult to convince readers of the formula’s validity. In addition, the limited number of experiments makes it hard to demonstrate the generalizability of the proposed method.

**Requested Changes:**

Please provide a detailed explanation of the motivation behind the improvements made to the OHD-Caps method, and clarify the underlying reasoning for proposing the F-CLIPScore calculation formula.
It is recommended to add more experiments to demonstrate the generalizability of the proposed method, and to present results on additional LLaVA-related benchmarks to show that the method does not negatively affect other benchmarks.
Furthermore, it is necessary to validate the effectiveness of the method on other more state-of-the-art LVLM models.

---

### Review · Reviewer_DLAr · 2025-10-20

**Summary Of Contributions:**

This paper addresses the problem of object hallucination in Large Vision-Language Models (LVLMs), where generated captions mention objects that are not present in the image. Prior work attributed this issue to the limited representational capacity of the vision encoder.
However, the authors argue that this explanation may be incomplete. They observe that even after fine-tuning, hallucination persists, and in some cases. This raises doubts about whether vision encoder capacity is the main cause.
To address this, the authors propose F-CLIPScore, an extension of CLIPScore that incorporates noun-level granularity by averaging the similarity scores of the full sentence and its constituent nouns. Without requiring additional training, they showed that F-CLIPScore improves hallucination detection accuracy (up to +39.6% ).
They further explore two use cases: (1) using F-CLIPScore as a training loss, which yields marginal gains over prior methods, and (2) using it as a data filtering metric to remove low-quality samples before LVLM training.
The authors conclude that F-CLIPScore is a valuable inference-time metric, and that object hallucination may stem more from limitations in evaluation and data quality than from vision encoder capacity itself. They also claim that using F CLIPScore as a data filtering tool during LVLM pretraining can reduce hallucination.

**Additional Comments:**

NA

**Audience:**

Yes

**Audience Explanation:**

The paper explores an interesting direction toward understanding the object hallucination issue in LVLMs from the perspective of evaluation metrics, though it still requires further development to support its claims with stronger evidence and analysis.

**Claims And Evidence:**

No

**Claims Explanation:**

I have several comments from different angles, which I point them out below.

------
### **Concerns on embedding distribution analysis (Figures 2b and 2c)**
The authors analyze cosine similarity distributions between pre-trained and fine-tuned CLIP models in Figures 2a–2c, to show that fine-tuning induces more substantial changes in text embeddings for hallucinated vs. non-hallucinated captions (Figure 2c), as compared to correct vs. incorrect predictions (Figure 2b). I have some concerns about the statistical rigor and interpretation on this experiment:

1. **Unequal statistical populations across figures:**
I have a question regarding the populations compared in Figures 2b and 2c; are they different in sample size? In 2b, each image contributes only one prediction (the caption selected by CLIP), whereas in 2c, all captions for each image (including both hallucinated and non-hallucinated) are considered? If so, then, this creates a mismatch in sample sizes and statistical properties, which could explain the apparent distributional differences in Figure 2c? This is to determine whether the observed shift is an effect of fine-tuning or an artifact of population sampling.

2. **Lack of formal statistical testing:**
The paper interprets distributional shifts visually (via histograms), but no statistical tests are provided to support claims about significant differences. I recommend including e.g. two-sample tests along with FDR correction to validate whether the observed shifts are statistically significant.

3. **Missing analysis of between-group distances:**
The authors focus on within-group similarity (i.e., how much each group’s embeddings changed after fine-tuning), but do not analyze the between-group separation, that is whether hallucinated and non-hallucinated captions become more distinguishable from each other after fine-tuning (similarly between correct and incorrect cases and similarity between hallucinated and non-hallucinated captions). This seems more relevant to the claim that fine-tuning helps discriminate hallucinations than the current reported distributions based on the similarity within the groups.

-----

### **Concerns about data filtering and validation of F-CLIPScore-based selection**

In Section 5.2, the authors apply F-CLIPScore as a data filter to remove the bottom x% of training samples from the LLaVA alignment set to show improved downstream performance on the POPE benchmark. I have several concerns regarding the interpretation and validation of this filtering procedure:

4. **Lack of verification of removed samples:**
The paper assumes that a low F-CLIPScore corresponds to “bad training examples” (e.g., hallucinated or misaligned image-caption pairs), but this is never verified. There is no inspection of the filtered-out examples to confirm their quality. Thus, the conclusion that these samples are undesirable is speculative.

5. **Assumption of proxy label without justification:**
F-CLIPScore is treated as a proxy for caption quality or hallucination, yet its reliability as such is not established beyond correlation with performance metrics. It remains unclear whether the metric accurately identifies hallucinated samples or simply filters out atypical and more challenging examples.

6. **Potential removal of challenging or rare samples:**
A critical concern I have is that F-CLIPScore may penalize semantically valid but rare, ambiguous, or challenging samples, which naturally score lower due to unusual content or composition. By removing these, the model may be trained only on easy or typical cases and overfits or reduces robustness. The observed performance gain might thus come at the cost of generalization.

7. **Implicit contradiction with dataset assumptions:**
The alignment dataset is originally constructed under the assumption that image-caption pairs are usable for training. Without evidence that removed samples are indeed faulty, the filtering contradicts this assumption without clear justification.

So, using F-CLIPScore as an automated system for data filtering could be interesting, but the paper’s interpretation of results should be significantly strengthened by a qualitative and quantitative analysis of the removed data. Also, measuring model performance on challenging or rare categories is necessary to assess whether the filtering introduces a bias toward “easy” examples.

---

### **Other comments:**

8. The paper frames its contribution around object hallucination in LVLMs, but the experiments are primarily focused on CLIP-based retrieval tasks. There is no evaluation of hallucination in generated captions, which is central to LVLM. This limits the strength of the claims regarding their method’s effectiveness in actual caption generation scenarios.

9. F-CLIPScore seems to be effective at inference. But its contribution as a training objective is very limited based on current results. So. this claim in the paper that it "**could further serve as an orthogonal method for training VLMs**" does not seem to be accurate to me.

10. In Table 1, I expect the authors to also report F CLIPScore values for the trained models and include standard deviations for the untrained cases for a fair comparison. Similarly, in Table 2, no standard deviations are provided, which makes it difficult to judge. As the CLIPScore (w/o training) already reaches 53.1, which is close to the best case of F CLIPScore (55.5), the absence of standard deviations prevents a meaningful statistical comparison between the methods.

**Requested Changes:**

Based on my comments, the following changes, clarifications, and improvements are required:

- clarification for statistical populations in embedding analyses

- add statistical tests (e.g. two-sample t-test) for embedding similarity distributions

- include analysis of between-group separation

- verify the quality of filtered-out samples

- assess the impact of filtering on challenging or rare samples

- reconsider the claim of orthogonality in training with FCLIPScore loss

- evaluate hallucination in caption generation

- report F-CLIPScore for trained models

- report standard deviations for all results

---

### Review · Reviewer_yfEB · 2025-10-22

**Summary Of Contributions:**

The paper presents a F-CLIPScore that is meant to better capture the image-text correlation than the traditional CLIPScore. The results are presented on a set of datasets and demonstrate a higher accuracy score when this metric is used for training. The paper is clearly written and well presented, but more insights could be given on the results to strengthen the quality of the findings.

**Audience:**

Yes

**Audience Explanation:**

The topic is of relevance to the TMLR audience encompassing technical development and new approaches to developing models and understanding hallucinations in vision captioning models.

**Broader Impact Concerns:**

No impact statement needed.

**Claims And Evidence:**

No

**Claims Explanation:**

The authors introduce the F-clip score which adds N additional scores to the original CLIPScore between the full caption embedding and the visual embedding by analysing also the noun-sub-part of the captions. While the idea is interesting and the paper is perfectly written and argumented, I find the results so far insufficient and would have appreciated a better coverage of the findings.

My main concern stems from the unseen bias that (1) encompasses. When we add N additional CLIP scores, we are essentially adding cosine similarities (not exactly as CLIPScore is w*max(cos(c,v),0)) but one can think in terms of proportionality) from a larger pool of vectors. This introduces a bias that comes just from sampling multiple times from the vector space. It is true that there is a normalization happening with N+1 at the denominator, but I am not sure that this is sufficient. In cases with semantically close terms (e.g. kitchen, stove, cupboard) the score is adding together terms that are coming from a dense region where they are likely to be close to each other (and hence have higher cosine similarity). This may be biasing the score higher not because the caption is better, but simply because there are more semantically close terms in it.
When we compare a short and a long caption that are equally correct this may become even more evident, e.g. "A mug in a kitchen cupboard next to a stove" could potentially score lower than "A mug or drinking cup tucked away in a kitchen cupboard or storage cabinet next to a stove, cooker, or oven" only because of the higher density of words in it.

One could also argue that the ad-hoc re-encoding of the words may not be the most robust way to obtain embeddings, as single-word inputs belong to a distribution that is shifted from what the text encoder was trained on. On the contrary, one could directly act at the attention level and use masks to obtain embeddings that only refer to the nouns in the sentence. The comparison between the two approaches would be interesting to see.

The result in Figure 2 is, to me, one of the most interesting of this paper, but it is poorly documented. It would have been interesting to see more insights on this aspect, for example presenting statistical tests that confirm this hypothesis first, on which relies the rest of the paper.

**Requested Changes:**

I would like to see more evaluations on:

- sequence length variations in the absolute character count and on the number of nouns
- two captions with the same semantic meaning but one augmented by synonyms for the nouns in it (so having different lengths)
- a better analysis on the claims associated to Figure2, for example by adding statistical tests on the two distributions and visualising  the distributions on cases with and without hallucinations (the ones in c but also for the image embeddings).
- In Table2 it would be good to see the distribution of removed samples, to verify if they are far from the ones kept during training (and if they actually contain hallucinations).

---

### Decision · Action_Editor_gXe7 · 2025-12-01

**Recommendation:** Accept with minor revision

**Audience:**

Yes

**Audience Explanation:**

The proposed F-CLIPScore achieves better performance than the original CLIPScore. From its formulation, it appears that explicitly incorporating noun information contributes significantly to the improvement. This simple yet effective modification is interesting and potentially valuable for the community.

However, (1) more clarification is helpful to emphasize the focus of retrieval-based setting as the central contribution of the paper in the current presentation. (2) As for the more trending topic on caption generation setting, this could be viewed as a seperated study in the future.

**Claims And Evidence:**

No

**Claims Explanation:**

All reviewers consistently highlight a central limitation: While contribution of F-CLIPScore alone is solid, the experiments focus exclusively on CLIP-based retrieval. This itself is supported by accurate evidence.  The current results primarily demonstrate benefits for retrieval-based contrastive models rather than improvements in caption generation.

A minor comment to revise the writing to avoid the potential confusion. Most readers expect object hallucination in vision–language models (VLMs) are studied in the settings caption generation, given their prior knowledge and current trends in the community. However, the study of this paper primarily focus on the discriminative settings. It is better to distinguish this two, and explicitly clarify the focus on this study, to close the gap between the potential expectation and real focus of this paper.